# Distribution, Effect, and Control of Exotic Plants in Republic of Korea

**DOI:** 10.3390/biology12060826

**Published:** 2023-06-06

**Authors:** Bong Soon Lim, Ji Eun Seok, Chi Hong Lim, Gyung Soon Kim, Hyun Chul Shin, Chang Seok Lee

**Affiliations:** 1Department of Bio & Environmental Technology, Seoul Women’s University, Seoul 01797, Republic of Korea; bs6238@swu.ac.kr (B.S.L.); jus826@swu.ac.kr (J.E.S.);; 2National Institute of Ecology, Seocheon 33657, Republic of Koreaothello1@nie.re.kr (H.C.S.)

**Keywords:** control, disturbance, ecological restoration, exotic plant, invasion

## Abstract

**Simple Summary:**

This study was carried out to clarify the spatial distribution of exotic plants at national, regional, and local levels, as well as their ecological impacts, and to prepare a strategy to reduce the impacts in Republic of Korea. A review of the biological characteristics of invasive plants showed that therophytes, annual plants, plants that disperse seeds by gravity (D4), erect form (E), and nonclonal growth form (R5) occupied the highest proportion. Exotic plants usually preferred disturbed areas such as lowlands, roadsides, and bare ground. At the national level, the distribution of exotic plants tended to be dominated by topographic conditions and increased around urbanized areas, agricultural fields, and coastal areas. At the regional level, they appeared in artificial plantations, vegetation due to disturbance, and vegetation established on lower slopes compared with upper slopes. At the local level, exotic plants appeared abundantly in the introduced vegetation, whereas they were rare in the native vegetation. Restorative treatments recovered species composition close to the reference vegetation and species diversity reduced by invasive species.

**Abstract:**

This study was carried out to clarify the spatial distribution of exotic plants at national, regional, and local levels, as well as their ecological impacts, and to prepare a strategy to reduce the impacts in Republic of Korea. This study was attempted at the national, regional, and local levels throughout Republic of Korea. Compositae occupied the highest percentage among invading exotic plants in Republic of Korea. A review of the biological attributes of exotic plants based on the dormancy form, longevity, disseminule form, growth form, and radicoid form showed that therophytes, annual plants, plants that disperse seeds by gravity (D_4_), erect form (E), and nonclonal growth form (R_5_) occupied the highest proportion. At the national level, the spatial distribution of exotic plants tended to depend on topographic conditions such as elevation and slope degree, and to increase around urbanized areas, agricultural fields, and coastal areas. The habitat types that exotic plants established were similar in their native habitat and in Korea, where they invaded. They preferred disturbed land such as roadsides, bare ground, agricultural fields, and so on. The spatial distribution of vegetation types dominated by exotic plants was restricted in the lowland. The proportion of the exotic/native plants tended to proportionate reversely to the vegetation type richness (the number of vegetation types); that is, the ecological diversity. The proportion of the exotic plants was higher in artificial plantations, vegetation due to disturbance, and vegetation established on lower slopes compared with upper slopes. Even at the local level, the exotic plants appeared abundantly in the introduced vegetation, while they were rare in the native ones. In the vegetation infected by exotic species, not only the species composition changed significantly, but the species diversity also decreased. Restorative treatment by introducing mantle vegetation around the hiking trail inhibited the establishment of exotic plants. Further, the restoration practice recovered the similarity of the species composition compared to the reference vegetation and increased the species diversity.

## 1. Introduction

The geographical distribution ranges of many species are restricted by the dispersal barriers of major environmental factors, including climate. As a result of geographical isolation, evolution proceeded in different patterns in each major area of the world [1,2,3]. Humans have greatly changed this pattern by transporting species throughout the world. Before industrialization, people moved cultivated plants and domestic animals from place to place when they established new agricultural fields and colonies. In modern times, a wide variety of species have been introduced, intentionally and accidentally, into places other than their native habitat [4,5,6,7,8,9,10].

Such exotic plants usually expand their distribution range beyond the place of initial establishment by leveraging their advantageous life-history strategies. In particular, the disturbed land provides microsites advantageous for exotic species equipped with opportunistic or ruderal life-history strategies [7,8,11,12,13]. Exotic plants with favorable life-history strategies in disturbed environments have reduced or replaced native species and changed ecosystem functions, raising concerns about the conservation of native ecosystems [7,8,13,14,15,16,17,18,19,20].

One of the major threats to biodiversity in the world is the direct destruction of their habitats by people through inadequate resource use or pollution. Another serious, but underestimated, problem is the threat to natural and seminatural habitats through the invasion of exotic species, which is a potentially lasting and pervasive threat [9,21,22,23,24]. When exploitation or pollution stops, the ecosystem often begins to recover; therefore, it is not a lasting threat. However, even if the introduction of alien organisms stops, the existing exotic species do not disappear; rather, they sometimes continue to spread and consolidate, and thus they are judged as a more serious threat [6,25,26,27].

Exotic species have the potential to invade other ecosystems and directly or indirectly affect native biota. They have, in fact, invaded all types of ecosystems on the planet, affecting their native biota. These species have been involved in many hundreds of extinctions, especially under islands conditions, whether it is on real islands or ecological islands. The environmental costs include the irreversible loss of native species and ecosystems [7,28,29,30]. 

Exotic plants can cause various problems in plant communities composed of native species: excluding native species; altering the habitat, hydrology, and nutrient cycle; significantly impacting biodiversity [8,15,31,32,33]. Exotic species can transform the structure and function of ecosystems by suppressing or excluding native species, either directly by competing for resources or indirectly by changing the way nutrients cycle through the system [28,33]. Increasing global domination by a few invasive species risks producing a homogenous world rather than a world characterized by great biological diversity and local distinctiveness [7,8,33].

The rate at which new invasions are detected is increasing over time across many ecosystems and regions. The observed rate is often increasing exponentially. Efforts to search for exotic species have increased in recent years, possibly contributing to the observed increase in invasions within some taxonomic groups. However, the overall pattern is consistent, arising from conspicuous and well-known taxa and in well-studied systems, indicating a dramatic increase in the invasion rates in the last half of the twentieth century [34,35,36,37,38].

Along with an apparent increase in the rate of invasions, the strong impacts of invasions have drawn widespread public and scientific attention, propelling policymaking and management actions. Therefore, many countries now have policies to prevent the risk of future invasions and to control the impact and spread of established non-native species [39,40,41].

The prevention of new invasions has a clear priority in new policies. Management measures against established invasions can also have the advantage of providing post-settlement options, but such efforts are often specific to the particular species and are potentially costly and long-term proposals [42,43,44]. The successful eradication of existing invasions is an impossible outcome in many instances, despite the significant improvements in the ability to detect new invasions early. Furthermore, a separate effort may be required for each invasion case, whether an eradication program or an effort to control the spread and abundance, even if it is the same species. In contrast, strategies for preventing new invasions target major transfer mechanisms, or vectors. These vector managements can be used to interfere with the transfer of a particular target species; at the same time, they are designed to prevent the wholesale transfer of diverse assemblages, including both target and non-target species, providing a robust and efficient management approach [7,28,36,45,46,47].

Invasive plants are typically controlled by applying physical or chemical methods [48,49], but they require repeated application [48]; despite that, the effect is not significant [50]. Despite the considerable time and resources required to develop and implement secure and effective eradication or control agents, these programs are often unsuccessful. Furthermore, without careful selection and execution, many chemical, mechanical, and biological control methods can be detrimental to non-target species and ecosystem health, which can facilitate the process of invasion. However, implementing restoration after the invasive species are eradicated or restoring ecosystems to their full potential are useful as a strategy to control invasive species [28,29,42,47,50,51,52,53,54,55].

Once the exotic species are established, the cost of management is dramatically increased and it is almost impossible to eradicate species that have begun to spread. Moreover, eradication can also create a bare ground, and thus promote reinvasion [56,57]. Therefore, management strategies should be prioritized over eradication measures. Restoration strategies reflecting ecological theory are based on limiting the assembly and invasion of exotic species [58,59,60,61].

The objective of this paper is to answer the following big questions of invasion ecology based on the results obtained from Republic of Korea: (1) Which taxa invade? (2) What types of ecosystems are vulnerable to invasive species and their impacts? (3) What is the ecological impact of their invasion? (4) How can we manage invasions?

To arrive at the goal, first of all we analyzed the family composition and the spatial distribution of the exotic plants at the national level. Secondly, we carried out a vegetation survey in eight major mountains of Seoul, and we analyzed the relationship between the occupancy ratio of the exotic plants and the landscape structure at the regional level based on the data. Thirdly, we analyzed the relationship between the performance of an exotic plant and the landscape structure on the mountain to clarify the environmental factors and the influences on the establishment and expansion of the exotic plant at the local level. Fourthly, we investigated the effects that the exotic plant caused based on species composition and diversity in a forest ecosystem and a riparian ecosystem. Finally, we evaluated the effect of restoration as a control measure of the exotic plant in a forest ecosystem and a riparian ecosystem.

## 2. Materials and Methods

### 2.1. Study Area

Study areas were selected at the national, regional, and local levels. The whole national territory was regarded as the study area at the national level. Seoul and Mt. Mido, which is located in the urban center of Seoul, were selected as the study areas at the regional and local levels, respectively. Korea is located in the middle latitudes of the Northern Hemisphere, Temperate Zone, and thus has four distinct seasons (between 38°21′ N, 126°11′ E and 33°12′ N, 129°52′ E; Figure 1). Annual mean temperatures in Korea range between 10 °C and 16 °C, except in the high mountain areas. The monthly mean temperatures range from 20 °C to 26 °C in the month of August, and from −5 °C to 5 °C in January. Annual precipitation is recorded as about 1500 mm and 1300 mm in the southern and the central parts of Korea, respectively. Winter precipitation is less than 10% of the total precipitation. Humidity reaches 70–80% nationwide in July, which is the highest. In contrast, monthly mean humidity remains at its lowest level in January and April, at 30–40% levels [62].

The Korean peninsula has experienced frequent human interferences as a corridor, which links between continent and sea. Moreover, Korea experienced the most rapid economic growth in the world and depends highly on trade. Those societal factors urge human disturbance as well. In addition, Korea also experienced severe artificial disturbance through wars, including the Second World War and the Korean War. All of those factors have contributed to the invasion and expansion of exotic species.

Seoul, the capital of Republic of Korea, is located in the central part of the Korean Peninsula and covers 605 km^2^ of land (126°46′15″ to 127°11′15″ E longitude, 37°25′50″ to 37°41′45″ N latitude; Figure 1). The mountainous vegetation of Seoul consists of five major plant communities distributed along an elevational gradient, with the *Pinus densiflora* community in the mountain peaks, the *Quercus mongolica* community in the upper slopes, the *Q. aliena* community in the lower slopes, and the *Zelkova serrata* and *Carpinus laxiflora* communities in the mountain valleys. *Alnus japonica* stands remain in the plains and valleys of the lowland that have escaped development, such as the cultural heritage areas [63,64,65]. Much of the natural landscape in the Seoul metropolitan area has disappeared due to extensive deforestation for fuel, building material, and other purposes during the 20th century [66]. The population of Seoul has increased from 2.4 million in 1960 to 9.8 million as of 2010 [67]. During this period, the proportion of green space has decreased from 70% in 1960 to 28% in 2017, mostly for housing [63,66,68]. The central government of Republic of Korea has designated most of the forested mountains located in the suburbs as green belts to prevent further loss of green space. Under the Seoul Metropolitan Government’s current Green Belt Ordinance, commercial, industrial, and urban development is not allowed in these forests [68].

Mt. Mido is located in Southern Seoul (Figure 1). Mt. Mido is an ecological island enclosed by a residential area (Banpo apartment complex), educational (Catholic University) and public facilities (National Library), a public facility (Court of Law), and a transportation facility (Gangnam Express Bus Terminal) in the east, west, south, and north, respectively. The vegetation of this site is composed of a plantation and secondary forest (Figure 1; [64]). The plantation is composed of *Robinia pseudoacacia*, *Populus tomentoglandulosa*, *Z. serrata*, and *Pinus koraiensis* plantations. The secondary forest is composed of *Q. acutissima*, and *Q. mongolica* communities. The vegetation of this site, especially the site facing the hiking trail, is severely affected by frequent visits by nearby residents, the indiscriminate introduction of sports and recreational facilities, and the indiscreet introduction of non-native plants for gardening and landscaping.

The effects of the exotic plants on the forest vegetation and the restorative treatment as a control plan of the exotic plants were investigated in Mt. Mido (Figure 1). The effect of the exotic plants on the riparian vegetation was investigated in the Han River watershed (Figure 1).

The effect of the restorative treatment as a control measure of the exotic plants was investigated in the Dongmun stream, which is located in Munsan-eub, Paju-si, Gyunggi province in central–western Korea. “Eub” and “Si” are the administrative units, which correspond to “town” and “city”, respectively. Although the riparian zone of the Dongmun stream was covered with natural vegetation, which were dominated by *Salix pierotii* (Korean willow), *S. gracilistyla*, and *Phragmites communis*, exotic plants, including *Ambrosia trifida* and *Sicyos angulatus*, are emerging as the new dominant species, as they have expanded their range rapidly in recent years [69,70]. The Suip stream, which is located in Bangsan-myun, Yanggu-gun, Gangwon province in eastern Korea (where “Myun” and “gun” are the administrative units, which correspond to “town” and “county”, respectively), was selected as the reference stream, as it was left to its natural process for more than 70 years after the Korean War.

### 2.2. Survey on Forest Vegetation

Distribution of exotic plants was investigated at three spatial scales, at the national (Republic of Korea), regional (Seoul, the capital of Republic of Korea), and local (Mt. Mido) levels. Ecological information for analysis on the nationwide distributional characteristics of the exotic plant species was obtained from the Ecobank of the National Institute of Ecology of Korea (http://www.nie-ecobank.kr, accessed on 30 May 2023). A field survey on distribution of exotic plant species was carried out based on the guideline for monitoring wild plants that disturb ecosystems [71]. Survey sites were designated at regular intervals throughout the whole national territory in Republic of Korea (refer to Figure 1).

The habitat types of each exotic plant were classified into the mountainous area, riparian zone, agricultural field, roadside, bare ground, and so on. Species nomenclature followed the Korean Plant Names Index [72]. The percentage of exotic plant species to native plant species was obtained based on flora survey data in each site. The field survey was conducted by recording the distribution area and coverage of the plants appearing in those quadrats after installing three quadrats (5 × 5 m^2^) at the locations selected in each grid. The ratio of exotic plants was calculated as the ratio of the number of exotic plants to the total number of plant species appearing in the target area.

We divided life-forms based on dormancy form, longevity, disseminule form, growth form, and radicoid form [73]. The dormancy form was divided into therophyte, wintering therophyte, hemicryptophyte, geophyte, chamaephyte, hydrophyte, and phanerophyte. The disseminule form was divided into disseminated widely by wind or water (D1), disseminated by attaching to or eaten by animals and man (D2), disseminated by mechanical propulsion of the dehiscence of fruits (D3), and having no special modification for dissemination (D4). The life-form based on longevity was divided into annual, wintering annual, biennial, perennial, and woody plants. The growth form was divided into the tussock form (T), branched form (B), procumbent form (P), erect form (E), partial-rosette form (Pr), and pseudo-rosette form (Ps). The radicoid form was divided into moderate extent of rhizomatous growth (R2), narrowest extent of rhizomatous growth (R3), clonal growth by stolons and struck roots (R4), and nonclonal growth (R5). An ecological map obtained from Seoul City (www.seoul.go.kr, accessed on 30 May 2023) was used to identify vegetation types and landscape boundaries in Seoul. Landscape ecological analyses of the maps were determined with the ArcView GIS software [74].

Vegetation data for study at the regional level were collected in Mts. Acha, Bukak, Buram, Cheonggye, Daemo, Gwanak, Inwang, and Surak, with five plots at each site in Seoul (Figure 1). A vegetation survey was conducted in forty plots, with five plots in each site [75].

The field survey on Mt. Mido selected for study at the local level was carried out from May to September 2002 and a resurvey was done from May to September 2017. A vegetation map (scale of 1:5000) of the study site was constructed with the GIS (Geographic Information System) program supported by ArcView [74] based on an urban ecological map [64] and field surveys. The distribution map of *Ageratina altissima* was prepared by representing the cover class of Braun–Blanquet [76], evaluated by the field survey on the maps at 1:5000 scales. Sections of the hiking trail were divided by expressing the homogeneous range of the cover class of *A. altissima*. The vegetation survey was conducted at 6, 21, and 10 plots in the *A. altissima*, *Q. mongolica*, and *Q. acutissima* stands, respectively, for a total of 37 plots.

The plot size was 20 m × 20 m in the studies of both regional and local levels. The vegetation survey was conducted by applying the phytosociological procedure of Braun–Blanquet [76]. Dominance of each species in each plot was evaluated on an ordinal scale, and each ordinal scale was converted to the median value of the percent of the cover range in each cover class. Relative coverage was considered as the importance value of each species. Relative percentage in percent was calculated by dividing the cover fraction of each species by the summed cover of all species in each plot, and then multiplying the value by 100. A matrix of importance values for all species in all plots was prepared and used as data for ordination using detrended correspondence analysis (DCA) [77]. To describe and compare species diversity and dominance among sites, rank–abundance curves [65,78,79] were prepared.

Restorative treatment was practiced by introducing *Rhododendron yedoense* var. *poukhanense* with a one m breadth beside the hiking trail to prevent the expansion of artificial disturbance on the trail.

### 2.3. Survey on the Riparian Vegetation

The cover class of all plant species in all plots installed randomly was recorded [76,80]. Plot sizes were 1 m × 1 m in the riparian zones dominated by herbaceous vegetation immediately adjacent to stream channels, 5 m × 5 m in the shrub lands, and 10 m × 10 m in the forests distant from the stream channels. Nomenclature followed the KNA (Korea National Arboretum) [72]. The cover class was estimated with the ordinal scale (from 1 for <1% to 5 for >75%) of Braun–Blanquet [76].

Experimental restoration was practiced by introducing *S. pierotii* and *S. gracilistyla* on the bank covered with *A. trifida* and *S. angulatus* with a 20 cm interval.

### 2.4. Statistics Analysis

We obtained Pearson’s correlation coefficient to deduce the relationship between the percentage of the exotic plants and environmental factors (α = 0.05). One-way analysis of variance (ANOVA) was used to compare the difference in the percentage of the exotic plants among the plant communities different in backgrounds and topographic locations (α = 0.05). The difference in values among the sites was tested by Scheffe’s test.

Data were analyzed using SPSS (version 24, SPSS Inc., Chicago, IL, USA) and R software version 4.2.3 (R Project for Statistical Computing).

## 3. Results

### 3.1. A Comparison among Taxa of Exotic Plants

Compositae occupied the highest percentage among exotic plants at 20.4% and followed in order by Gramineae (19.9%), Cruciferae (9.0%), Leguminosae (7.0%), Solanaceae (4.2%), Caryophyllaceae (3.4%), Amaranthaceae (3.1%), Polygonaceae (2.8%), Scrophulariaceae (2.8%), Malvaceae (2.5%), Convolvulaceae (2.5%), Chenopodiaceae (2.0%), Umbelliferae (2.0%), and so on (Figure 2, Appendix A).

### 3.2. Life-Form Composition of Exotic Plants

In the dormancy form composition of the invading exotic plants in Korea, therophytes occupied the highest percentage, followed by wintering therophytes, hemicryptophytes, geophytes, chamaephytes, hydrophytes, and phanerophytes (Figure 3). Based on the longevity, annual plants occupied more than half, followed by perennial, biennial, wintering annual, and woody plants (Figure 3). The percentage of the disseminule form of the invading exotic plants in Korea was higher in the order of dispersal by gravity (D4), dispersal by wind and water (D1), dispersal by animal (D2), and dispersal by elasticity (D3) (Figure 3). The percentage of the growth form was higher in the order of the erect form (E), procumbent form (P), branched form (B), tussock form (T), partial-rosette form (Pr), and pseudo-rosette form (Ps). The radicoid form of the invading exotic plants in Korea was higher in the order of the R5, plants isolated without any connecting organ, R3 with a narrow rhizome, R4 with a stolon, and R2 with a relatively wide rhizome (Figure 3).

### 3.3. Habitat Types of Exotic Plants

Habitat types in the original habitats of exotic plants that established in Korea showed a higher frequency in roadsides, bare ground, agricultural fields, pastures, riversides, polluted soils, wetlands, and landfill areas (Figure 4). Habitat types invaded in Korea showed a trend similar to that in their original habitats, as the percentage followed the order of roadsides, bare ground, seashores, landfill areas, pastures, wetlands, agricultural fields, forest edges, and riversides (Figure 4).

### 3.4. Spatial Distribution of Exotic Plants at the National Level

Spatial distribution of exotic plants, elevation, slope degree, percentage of urbanized land, and percentage of agricultural fields throughout the whole national territory of Republic of Korea is depicted in the grid maps (Figure 5). The occupied rate of the exotic plants tended to depend on the elevation and slope degree, and to increase around urbanized areas, agricultural fields, and coastal areas (Figure 5).

As a result of the correlation analysis, the percentage of exotic plants showed significant correlation with the elevation (negative logarithmic), slope degree (negative linear), the percentage of urbanized land (second function), and the percentage of agricultural fields (linear) (Figure 6).

### 3.5. Spatial Distribution of Exotic Plants at the Regional Level

The spatial distribution of plant communities dominated by exotic plants at the regional level showed that their distribution was restricted to lowlands compared to uplands (Figure 7).

The percentage of exotic to native plants investigated in eight major mountains tended to proportionate reversely to the vegetation type richness (Figure 8).

Comparing the percentages of the exotic plants among plant communities with different traits, the ratio tended to be higher following the order of artificial plantations (constructed by introducing *Alnus hirsuta*, *Alnus hirsuta* var. *sibirica*, *R. pseudoacacia*, *P. tomentiglandulosa*, *Pinus rigida*, *P. koraiensis*, *Larix leptolepis*, and *Betula platyphylla* var. *japonica*, respectively), vegetation due to disturbance (dominated by *Sorbus alnifolia*, *Arudinella hirsuta*, and *Lespedeza cyrtobotrya*, respectively), vegetation established in lower slopes (dominated by *A. japonica*, *Prunus sargenntii*, *Q. acutissima*, *Fraxinus mandshurica*, and *P. densiflora*, respectively), and natural vegetation established in upper slopes (dominated by *Q. mongolica*, *Betula davurica*, *Q. variabilis*, *P. densiflora*, *Q. serrata*, *C. laxiflora*, *Q. aliena*, and *Z. serrata*, respectively) (Figure 9).

### 3.6. Spatial Distribution of A. altissima as an Exotic Plant

In 18 sections (Figure 1), divided by a homogeneous range of cover classes, cover classes of *A. altissima* ranged from none to IV. The cover class of *A. altissima* was the highest in sections 1 and 14 as cover class IV, and sections 2 and 13 (cover class III), section 11 (cover class II), followed by sections 3, 10, 12, and 15 (cover class I). On the other hand, *A. altissima* did not appear in sections 5, 6, 7, 8, and 9, which are located around the summit, and sections 16 and 17, which are located amidst young Korean pine plantations afforested recently (Figure 10).

### 3.7. The Ecological Effects of Exotic Plant Infection

The effects of the exotic plant infection were investigated in terms of changes of species composition and species diversity. The effect in the forest ecosystem was investigated by comparing the species composition and diversity of a plant community dominated by *A. altissima* with three native plant communities dominated by *Q. mongolica* and *Q*. *acutissima*.

As a result of the DCA ordination based on the vegetation data, the species composition of stands infected with a tree-of-heaven showed a large difference from that of Mongolian oak stands, which was established on the upper slope. Although it was not a large difference, it also showed a difference in species composition from stands of *Q*. *acutissima*, which were established on the mid-slopes (Figure 11).

The effects of the exotic plant infection were also investigated in terms of changes in species composition and diversity in the riparian ecosystem. The effect on the riparian ecosystem was investigated by comparing the species composition and diversity of two plant communities dominated by *A. trifida* and *S. angulata* with those of three native plant communities dominated by *S. pierotii*, *S. gracilistyla*, and *Phragmites japonica*, which represent the vegetation zones dominated by herbaceous plant, shrub, and tree in the riparian zone.

As the result of the DCA ordination based on the vegetation data, the species composition of the stands infected with two exotic plants was remarkably different from those of *Phragmites japonica* (herb), *S. gracilistyla* (shrub), and *S. pierotii* (tree), which dominate the native riparian vegetation in Republic of Korea (Figure 12).

Species diversity was compared by the species rank–abundance curve, and the richness of the stands infected with a tree-of-heaven was lower than those of the three oak communities, and the slope of the curve that the exotic plant community formed was steeper than those done by the native plant communities (Figure 13).

The richness of the stands infected with two exotic plants established in the riparian zone was lower than those of three native riparian vegetation stands, and the slope of the curves that the two exotic plants communities formed were steeper than those done by the native riparian plant communities (Figure 14).

### 3.8. The Effects of Ecological Restoration for Control of Exotic Plant Species

Restorative treatment by introducing mantle vegetation around the hiking trail reduced the cover of *A. altissima* in sections 1, 2, 11, 13, and 14. However, the cover increased in sections 5, 7, 16, 17, and 18 despite the treatment. On the other hand, sections 3, 4, 6, 8, 9, 10, 12, and 15 did not show any change (Figure 10).

As the result of the DCA ordination based on the vegetation data obtained from the nonrestored stand, which were dominated by *A. trifida* and *S. angulate*, the sites were restored by imitating the natural river dominated by *S. pierotii*; in the reference river, the nonrestored stands showed a species composition remarkably different from that of the reference stands, while the restored sites showed a species composition similar to that of the reference stands (Figure 15).

The richness of the restored sites was higher than that of the reference stands. The richness of the nonrestored stands was the same as that of the reference, but the slope of the curves that the former stand formed was steeper than that of the reference stand (Figure 16).

## 4. Discussion

Ecologists studying exotic species try to address the following basic questions: Which taxa often invade? How fast do they invade? What kinds of ecosystems are vulnerable to exotic taxa and their impacts? What are the ecological effects of their invasion? How can we contain, control, or eradicate harmful invaders? Our discussion was focused on those five questions.

### 4.1. Which Taxa Invade?

Among 357 exotic plants investigated in Korea, compositae occupied the highest percentage among exotic plants at 20.4%, followed in order by Gramineae (19.9%), Cruciferae (9.0%), Leguminosae (7.0%), Solanaceae (4.2%), Caryophyllaceae (3.4%), Amaranthaceae (3.1%), and so on (Figure 2, [81,82]). On the other hand, reviewing the biological attributes of exotic plants based on the dormancy form, longevity, disseminule form, growth form, and radicoid form showed that therophytes, annual plants, plants that disperse seeds by gravity (D4), erect forms (E), and R_5_ forms occupied the highest proportion (Figure 3).

Taxa, which account for a higher percentage, usually have better dispersal agents, higher reproductive capacity, and shorter life cycles [83]. McNeeley [28] suggested the following five predictions regarding invasive species: Firstly, the probability of a species becoming invasive increases as the initial population size increases, so species that are introduced intentionally and kept under cultivated or maintained under animal husbandry over a long period are more likely to establish. Secondly, species with larger native geographic ranges are more likely to become invasive than those with smaller native ranges. Thirdly, species invading a country or region should be considered as a high risk of becoming invasive in an ecologically or climatically similar country or region. Fourthly, species with specialized pollinators are less likely to become invasive unless they are introduced together. Finally, successful invasions usually require that the new habitat conditions are similar to those at the origin, especially in terms of climate conditions. Another group of exotic species are those that have expanded their ranges within the continental areas because they fit the ways in which humans have altered the environment [84]. The other special class of exotic species includes those that have close relatives in the native biota. When exotic species hybridize with indigenous species and varieties, unique genotypes may be removed from local populations and taxonomic boundaries may become obscured [85].

The results of this study, showing that compositae account for the highest proportion of invading taxa in Korea (Figure 2), with the life-form composition of therophytes, annual plants, plants that disperse seeds by gravity (D4), erect forms (E), and R_5_ forms occupying the highest proportion (Figure 3), are well in line with the general characteristics of the exotic plants mentioned before.

### 4.2. How Fast Do They Invade?

The rate of spread is a function of both reproduction and dispersal; there are species that reproduce quickly and spread much faster [86]. In order to determine the rate of spread of plants, information on rare dispersal events that can send plants over an unusually long distance is required. While the rate of dispersal is critical, other factors, such as reproductive maturity age, disturbance frequency, intensity of habitat disturbance, and fecundity, are also important. Seeds can often be transported over long distances by agents such as water, wind, vehicles, or livestock at very high speeds [28].

Among the invading exotic plant species in Korea, therophytes and annular plants accounted for a high proportion, and the disseminule form of seeds was high in plants that disperse seeds by gravity (D4) (Figure 3). These biological attributes correspond to favorable conditions to reproduce quickly and spread more rapidly [86].

On the other hand, as the type of habitat where the exotic plant species established included roadsides, bare ground, and coastal areas, accounting for a high proportion, the result was similar to the origin of the habitat types of those plant species (Figure 4). These results mean that such exotic plants are properly finding their preferred places, which is interpreted as a result of active human and material exchanges around the world. In reality, the increased mobility of people and their goods increases the likelihood of moving species around the world [27,28].

### 4.3. What Types of Ecosystems Are Susceptible to Exotic Plant Species and Their Impacts?

One reason that exotic species can so easily invade and dominate new habitats and replace native species is that there are no natural predators, pests, and parasites in the new habitat. Human activity may create unusual environmental conditions, such as nutrient pulses, increased disturbances including fire, or increased light intensity, to which exotic species can adapt more easily than native species. The higher occurrence of exotics is often found in habitats that have been further altered by human interference. Fragmented forests, suburban developments, and easy access to landfills have allowed for an increase in the numbers and ranges of so-called wandering species [12,33,44,87].

All ecosystems, including those in well-protected national parks, could potentially be invaded, but some seem more vulnerable than others. Evolutionarily and geographically isolated ecosystems, especially oceanic islands, are particularly vulnerable. Urban–industrial areas, habitats suffering from periodic disturbance, ports, lagoons, estuaries, and water fronts, where the effects of natural and artificial disturbances often coexist, are also particularly vulnerable to invasions [88,89]. Virtually all ecological communities are vulnerable to invasion to some extent, and artificial disturbance increases the vulnerability of most ecosystems. Therefore, the continued expansion of human activity is likely to increase the vulnerability of ecological communities to invasion [28,44,90].

In the result of this study, exotic species tended to be distributed in places with low elevations and gentle slopes, around cities and agricultural fields, and along the coastal area at the nation level (Figure 5). Therefore, the percentage of exotic species tended to be proportionate to the elevation and slope degree (negatively), and the urbanization ratio and percentage of agricultural fields (positively) (Figure 6). The vegetation type dominated by exotic plant species investigated in Seoul was restricted to the lowlands, with frequent disturbances (Figure 7). The proportion of exotic plant species surveyed in eight mountains within Seoul city showed the opposite trend to the richness of vegetation types that those mountains possessed (Figure 8). The low richness of vegetation types means that the lowlands of the mountain have been transformed into urbanized areas, including residential areas. As a result of comparing the ratio of exotic plant species by vegetation type, exotic species showed a high rate in the artificial afforestation and the vegetation type caused by disturbance, whereas they appeared at a low rate in natural vegetation (Figure 9). In an isolated mountain surrounded by urbanized areas, the coverage of *A. altissima* surveyed along the hiking trail was high along the low-lying hiking trail close to the residential area, and it was low or did not appear around the mountain peak dominated by natural vegetation (Figure 10). These results are evidence to demonstrate that the invasion of exotic plants begins at sites disturbed by human influence, and that the spread is also due to the effects of such disturbances.

The forest edge, including the hiking trail, is vulnerable to the invasion of exotic plants due to physical disturbance and nutrient input, as a place where organisms, matter, and energy are exchanged between the two habitats [91,92,93,94]. Thus, exotic plants are abundant while species diversity is lower in the forest edge compared with the undisturbed forest interior [95,96,97,98]. In reality, experimental manipulation, which removed the forest canopy and undergrowth, facilitated the invasion of exotic plants [99,100]. This phenomenon suggests that the invasion and expansion of exotic plants are closely related to artificial disturbances. The results of this study resemble the facts commonly known regarding the invasion and expansion of the exotic plants.

*A. altissima*, as an exotic plant species, appears more abundantly at the lower elevations, where frequent artificial interferences are prevailing, than at the higher elevations, where such impacts are lessened (Figure 10). On the other hand, they appeared abundantly in the introduced vegetation, such as the black locust plantation, but they did not appear or were rare in the natural one, such as the oak forest (Figure 10). However, an exceptional phenomenon was found at an entrance C. The lack of *A. altissima* in the entrance was due to the dense coverage of the recently afforested Korean pine stand (Figure 10). In this respect, the light condition is also important [65].

### 4.4. What Is the Ecological Impact of Exotic Species Invasion?

Every exotic species alters the species composition of native biological communities in some way. Whether it becomes invasive, and thus harmful, depends on the specific nature of the exotic species, the vulnerability of the host ecosystem, and opportunity [101,102,103,104]. Changes in the ecosystems may be initiated by natural disturbances, such as storms, earthquakes, volcanic eruptions, fires, climate, or management regimes, but are reinforced or accelerated by the invasion of exotic species. Land transformation and invasions are interlinked, bringing more opportunities for invasion [27,33,105,106].

The species composition of an ecosystem at any given location and time is determined depending on the current environmental conditions, the levels and types of disturbance, the balance of loss and recruitment, and the composition of the regional species pool. Increasing levels of human-induced ecosystem transformation may accelerate environmental change, and the dramatic increase in the intentional and accidental biota transport across the world inevitably will increase the regional species pool, whereas it will perhaps also decrease native species and ultimately decrease the global species pool. The combination of these factors lays the base for a radical alteration of an ecosystem [12,28,107,108,109,110].

In a result of this study, the species composition of vegetation types infected with a tree-of-heaven showed a difference from that in the reference area, where the exotic species did not invade (Figure 11). The species composition of riparian vegetation infected with giant ragweed and *S. angulatus* also changed significantly compared with that in the reference area, where they did not invade (Figure 12). In both forest ecosystems and riverside ecosystems, species diversity of vegetation types in which exotic species invaded decreased (Figure 13 and Figure 14).

### 4.5. Restoration as a Tool to Inhibit Invasion and Expansion of Invasive Species

Invasive species provide a serious challenge to environmental managers because of their explosive growth [111,112]. It is known that disturbed ecosystems are usually more vulnerable to the infestation of exotic plants than undisturbed ecosystems [44,87,113]. The results of this study showed that the invasion and spread of exotic species were closely related to disturbance at national, regional, and local levels. In this respect, it is imperative to protect disturbed ecosystems from human interference, such as excessive use and management, to prevent the invasion and expansion of exotic species. It is imperative to foster closed undisturbed conditions to discourage disturbance-adapted exotic plants [44,114]. In fact, a comprehensive restoration of entire ecosystems may be necessary [115,116]. Artificial interference for forests is currently declining in rural areas in Korea, but interference due to forest management still remains in urban areas, which may cause further invasion of exotic plants. Therefore, we propose a management plan with ecological restoration principles reflected to address ecosystems infected with exotic species [30,117,118].

How do we return a disturbed ecosystem to a stable ecosystem? Can we contain, control, or eradicate harmful invaders through such restoration? The goal of restoration ecology is to aid the recovery of an ecosystem that has been disturbed or damaged by external influences, such as fire, logging, mining, agriculture, or urban development. A major goal of restoration practitioners is to return a habitat to a more desirable condition, involving a particular species composition, community structure, and/or set of ecosystem functions [119,120,121].

Restoration is used to maintain the overall health and sustainability of an ecosystem, and thus can be the most effective way to increase the resilience and resistance of an ecosystem to invasion by exotic species. Although restoration is used for various purposes, it is particularly being used as a way to combat exotic species, especially under the stresses of climate change and the expanding influence of human activities. Optimal, balanced, and stable ecological conditions arising as a result of restoration can more effectively limit the spread and settlement of exotic species [50,122,123,124].

In a result of this study, the restorative treatment that introduced mantle vegetation around the hiking trail where *A. altissima* invaded reduced the coverage of the exotic plant (Figure 10). In addition, the riparian vegetation, where giant ragweed invaded, showed significantly different species composition from the reference vegetation and showed low species diversity (Figure 12 and Figure 14). However, the restoration implemented by introducing Korean willow restored species composition similar to the reference vegetation and increased species diversity (Figure 15 and Figure 16). This result suggests that plantings of willow could help restore riparian zones that have been overtaken by giant ragweed. In reality, willows can act as a nurse crop for other understory plants by ameliorating high light, temperature, and soil moisture [125,126]. In particular, the decline of giant ragweed dominance by willow shading might increase plant diversity in this restored riparian ecosystem. Based on the results, we suggest a comprehensive restoration plan which recovers the integrity of the ecosystems disturbed due to various human activities to inhibit the invasion and expansion of exotic species [29,95,127,128].

## 5. Conclusions

Compositae occupied the highest percentage among exotic plants, followed by Gramineae, Cruciferae, Leguminosae, and so on. At the national level, the spatial distribution of exotic plants tended to depend on topographic conditions, such as the elevation and slope degree, and to increase around urbanized areas, agricultural fields, and coastal areas. The habitat types that exotic plants established were similar in their native habitat and in Korea, where they invaded. They preferred disturbed lands such as roadsides, bare ground, agricultural fields, and so on. The spatial distribution of vegetation types dominated by exotic plants was restricted in the lowlands. The proportion of the exotic/native plants tended to proportionate reversely to the vegetation type richness; that is, ecological diversity. The proportion of the exotic plants was higher in the artificial plantations, vegetation due to disturbance, and vegetation established in lower slopes compared with the upper slopes. Even at the local level, exotic plants appeared abundantly in the introduced vegetation, whereas they were rare in the native ones. In the vegetation infected by exotic species, not only the species composition changed significantly, but also the species diversity decreased. Synthesizing these results, the invasion of exotic plants begins at sites disturbed by human influence, and the spread is also due to the effects of such disturbances. Restorative treatment by introducing mantle vegetation around the hiking trail inhibited the establishment of exotic plants. Further, the restoration practice recovered the similarity of the species composition compared to the reference vegetation and increased the species diversity. Based on the results, we suggest a comprehensive restoration plan which recovers the integrity of the ecosystems disturbed due to various human activities to inhibit the invasion and expansion of exotic species.

## Figures and Tables

**Figure 1 biology-12-00826-f001:**
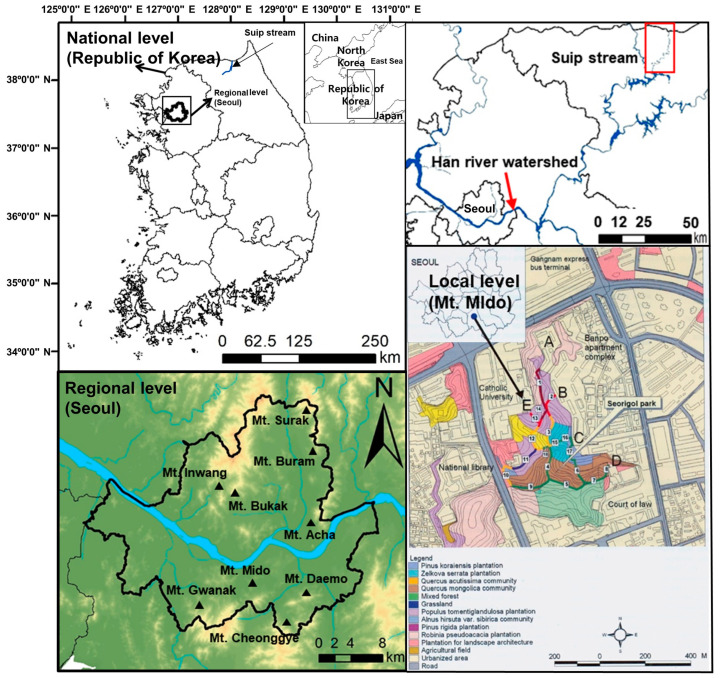
Maps showing the study areas at the national (**upper left**), regional (**lower left** and **upper right**), and local (**lower right**) levels. A map showing a study area at the local level represents the spatial distribution of vegetation and the cover of *Ageratina altissima* in the study area, Mt. Mido located in southern Seoul. A, B, C, D, and E indicate the entrances of the urban park and numbers in the quadrangle show the section number of the hiking trail.

**Figure 2 biology-12-00826-f002:**
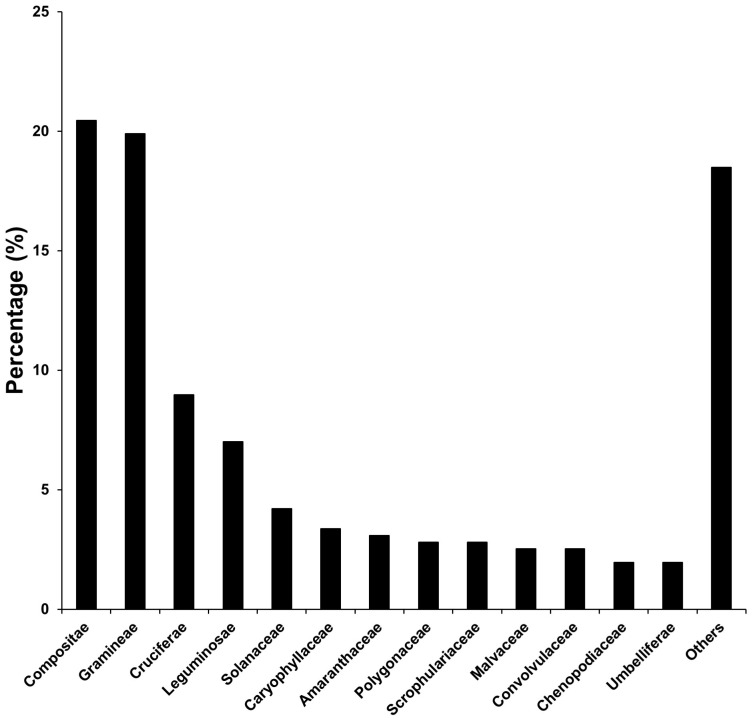
Proportion of exotic plants by family found in Republic of Korea.

**Figure 3 biology-12-00826-f003:**
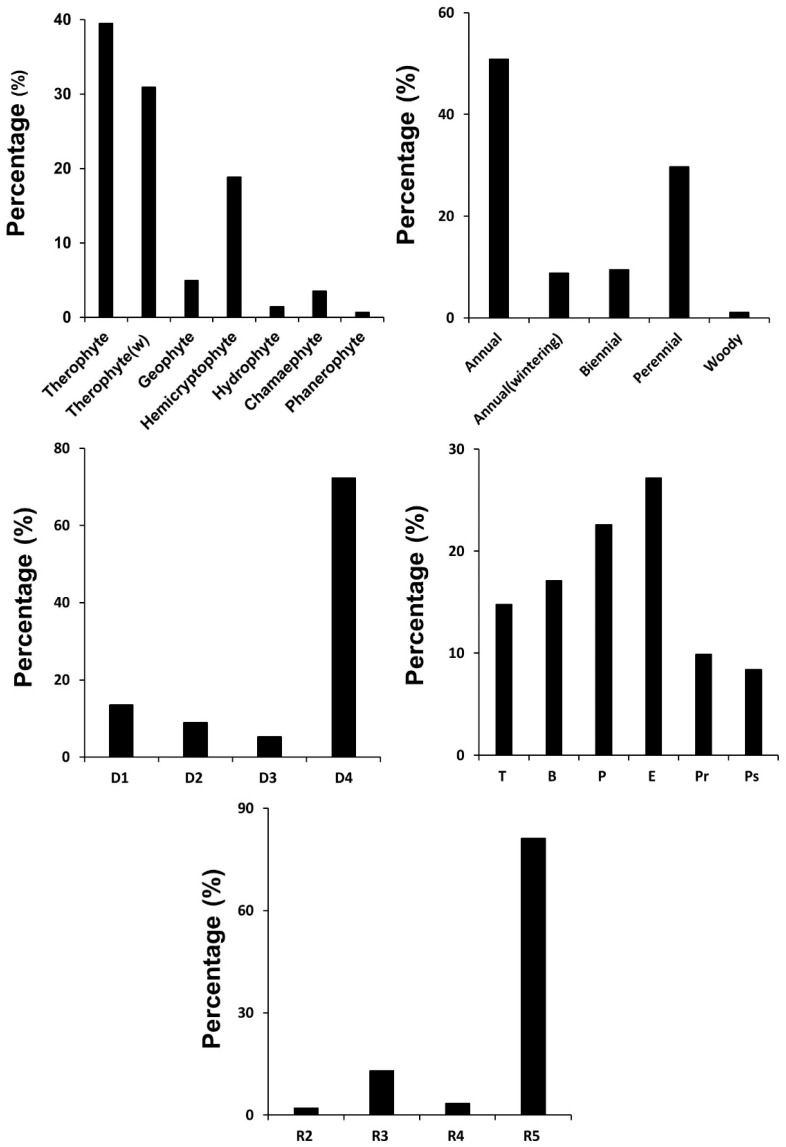
Dormancy form composition of the invading exotic plants in Korea. D1: disseminated widely by wind or water; D2: disseminated attaching to or eaten by animals and man; D3: disseminated by the mechanical propulsion of the dehiscence of fruits; D4: having no special modification for dissemination. T: tussock form; B: branched form; P: procumbent form; E: erect form; Pr; partial-rosette form; Ps: pseudo-rosette form; R2: moderate extent of rhizomatous growth; R3: narrowest extent of rhizomatous growth; R4: clonal growth by stolons and struck roots; R5: nonclonal growth.

**Figure 4 biology-12-00826-f004:**
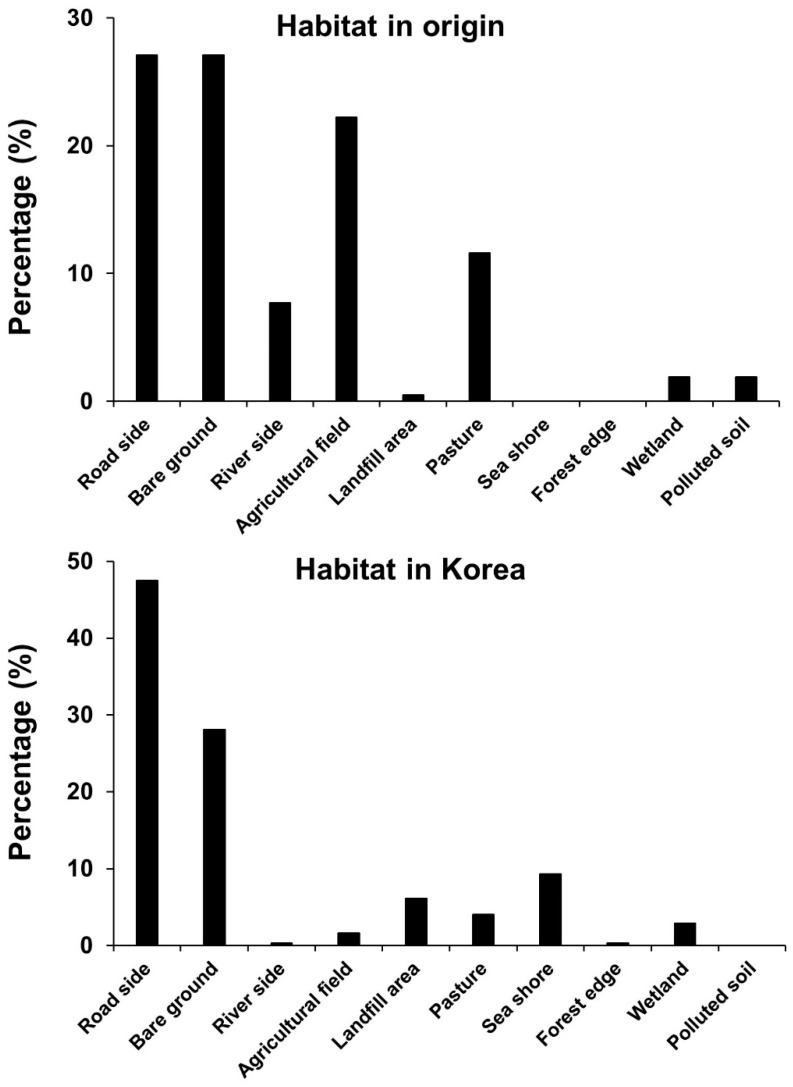
Habitat type composition of the invading exotic plants in Korea in their original place (**upper**) and Korea (**lower**).

**Figure 5 biology-12-00826-f005:**
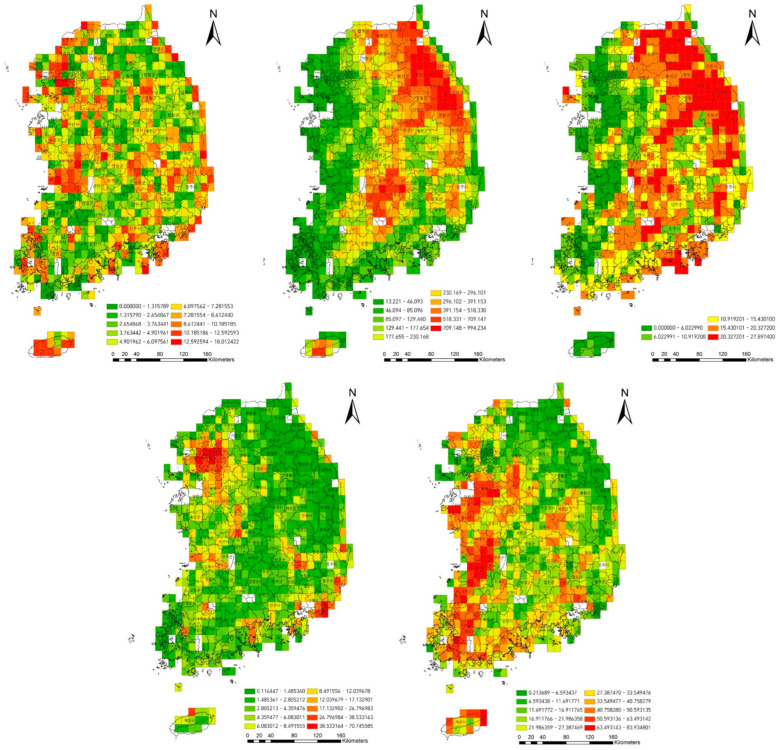
Spatial distribution of percentage of the exotic plant species (**upper left**), elevation (**upper center**), slope degree (**upper right**), urbanized rate (**lower left**), and percentage of agricultural fields (**lower right**) throughout the whole national territory of Republic of Korea. Korean letters (“-군”, “-시”) on maps are the administrative units, which correspond to “county” and “city”, respectively.

**Figure 6 biology-12-00826-f006:**
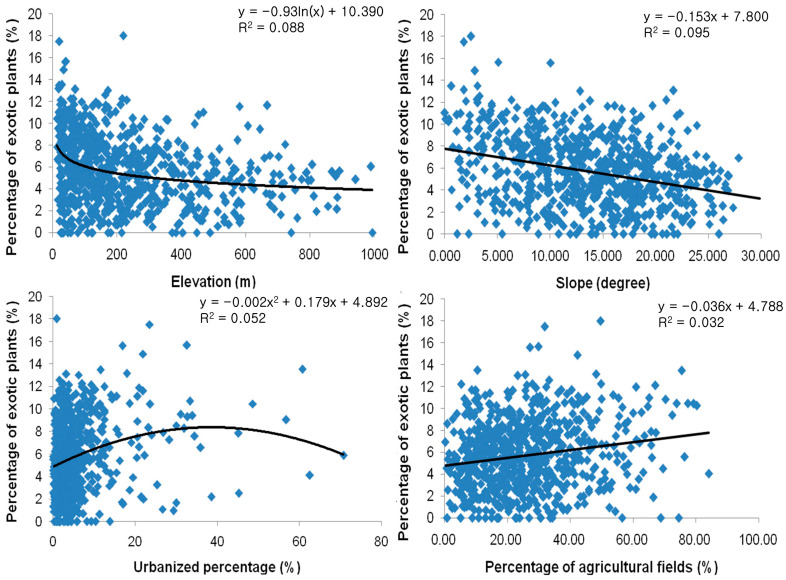
Relationships between elevation, slope, percentage of urbanized land, and percentage of agricultural fields and percentage of exotic plants (α = 0.05).

**Figure 7 biology-12-00826-f007:**
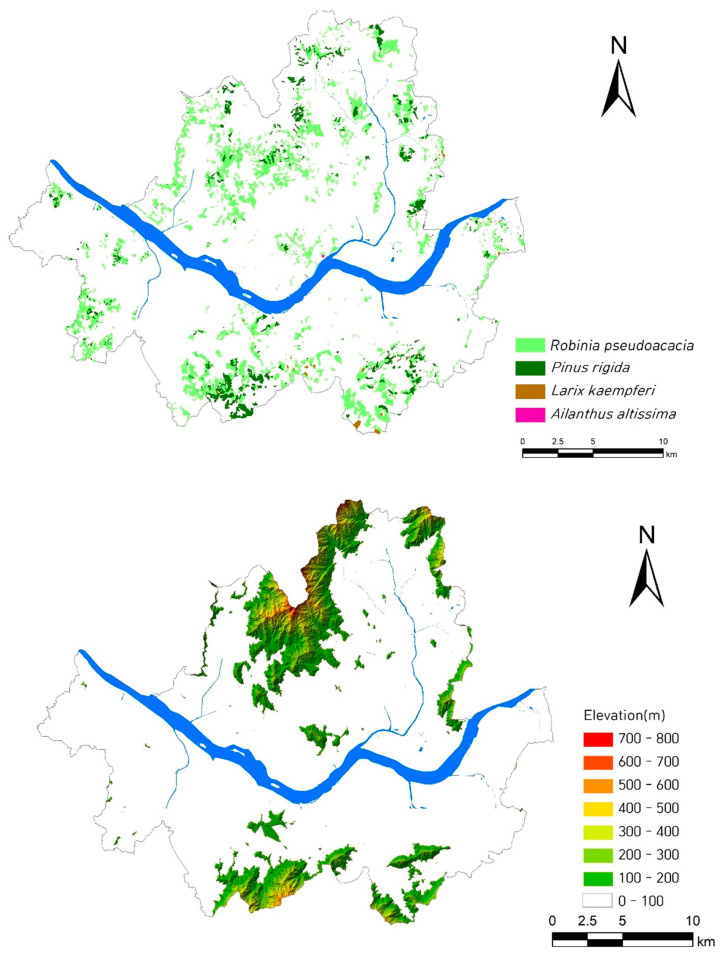
Spatial distribution of plant communities dominated by exotic plants (**upper**) and elevation (**lower**) in the Seoul Metropolitan Area, central–western Korea. Their distribution is restricted to the lowlands.

**Figure 8 biology-12-00826-f008:**
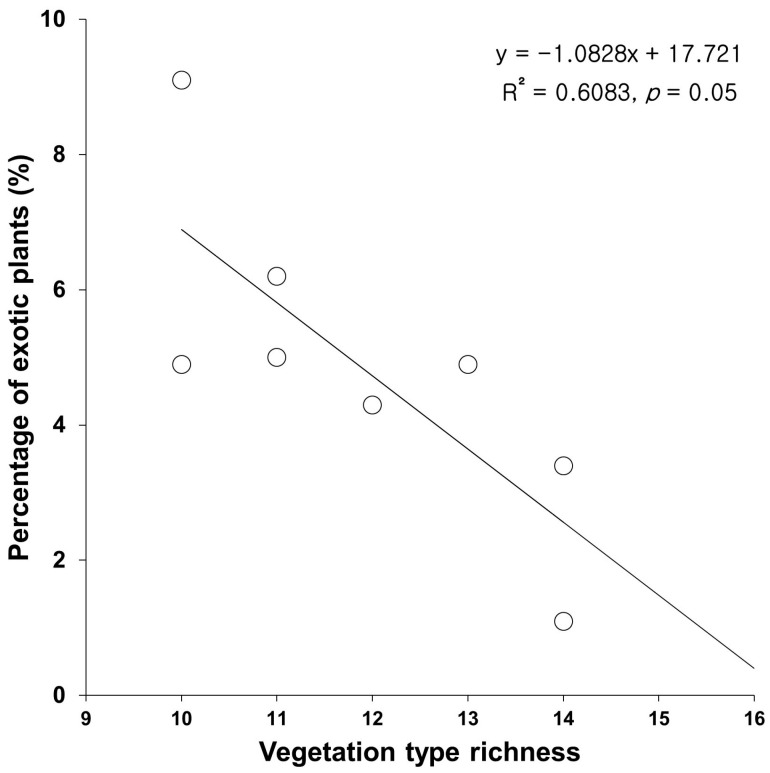
Relationship between percentage of exotic plants and the number of vegetation types in eight major mountains of Seoul.

**Figure 9 biology-12-00826-f009:**
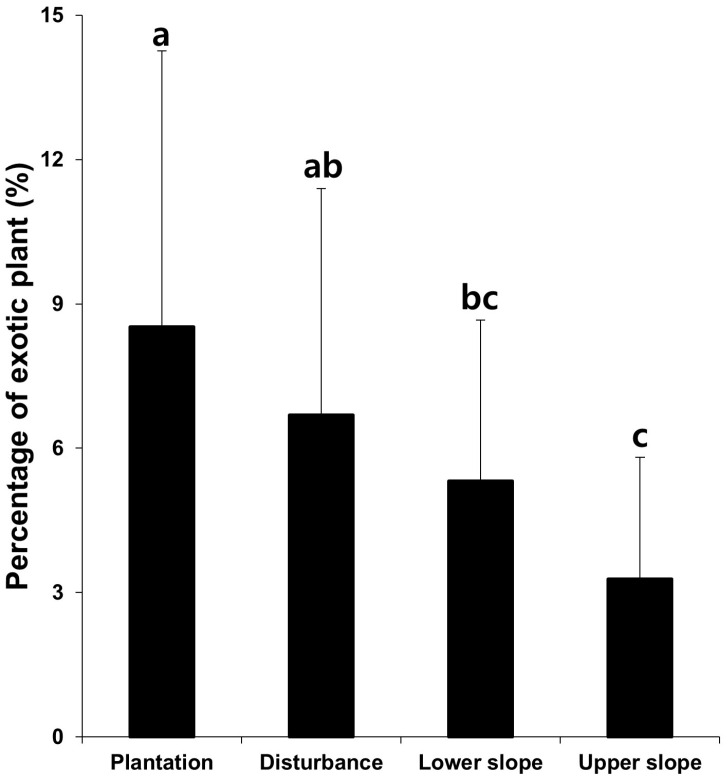
A comparison of the percentage of exotic plant species among plant communities with different traits in eight major mountains of Seoul. Each bar expresses the mean and standard deviation of the mean. An ANOVA test was conducted on the percentages of the exotic plants in the plant communities different in establishment backgrounds and topographic locations at α = 0.05; the means with the same alphabetical character (in superscript) for each parameter were not different from each other. Error bars indicate the standard deviation of the mean percentage of exotic plants for each vegetation type. Disturbance: plant communities due to disturbance; Lower slope: plant communities established in lower slopes; Upper slope: plant communities established in upper slopes.

**Figure 10 biology-12-00826-f010:**
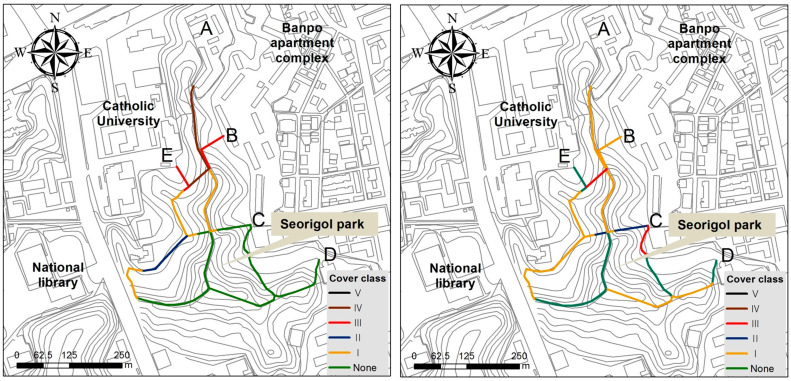
Maps showing the cover class of an exotic plant, *A. altissima*, before (**left**) and after (**right**) a restorative treatment, which was implemented by introducing the Korean azalea beside the hiking trail. The section numbers of the hiking trail were shown in Figure 1. A, B, C, D, and E indicate the entrances of the urban park.

**Figure 11 biology-12-00826-f011:**
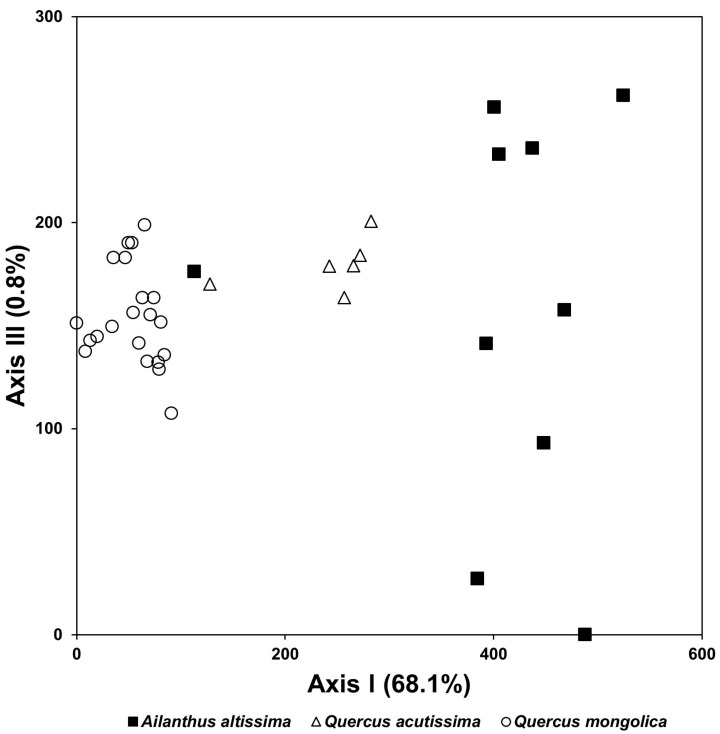
DCA ordination of stands based on vegetation data collected from native oak (*Q*. *acutissima* and *Q. mongolica*) stands and stands infected with an exotic plant, tree-of-heaven, based on vegetation data.

**Figure 12 biology-12-00826-f012:**
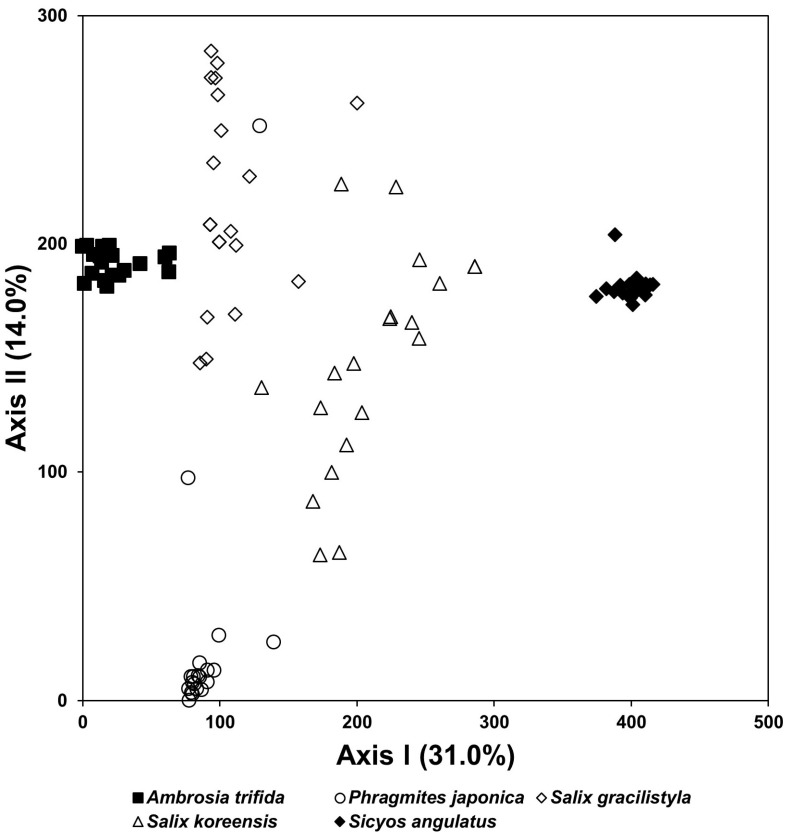
DCA ordination of stands based on vegetation data collected from three natural vegetation types dominated by *S. pierotii*, *S. gracilistyla*, and *P. japonica* and two vegetation types infected with two exotic plants, *A. trifida* and *S. angulata*.

**Figure 13 biology-12-00826-f013:**
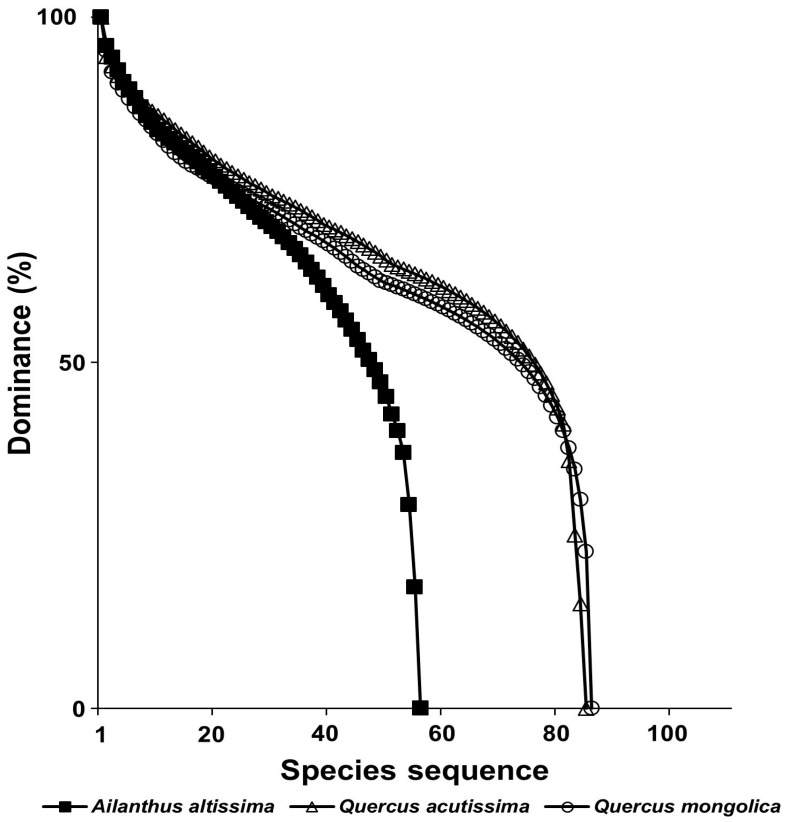
A comparison of species rank–abundance curves of *A. altissima*, *Q. acutissima*, and *Q. mongolica* stands.

**Figure 14 biology-12-00826-f014:**
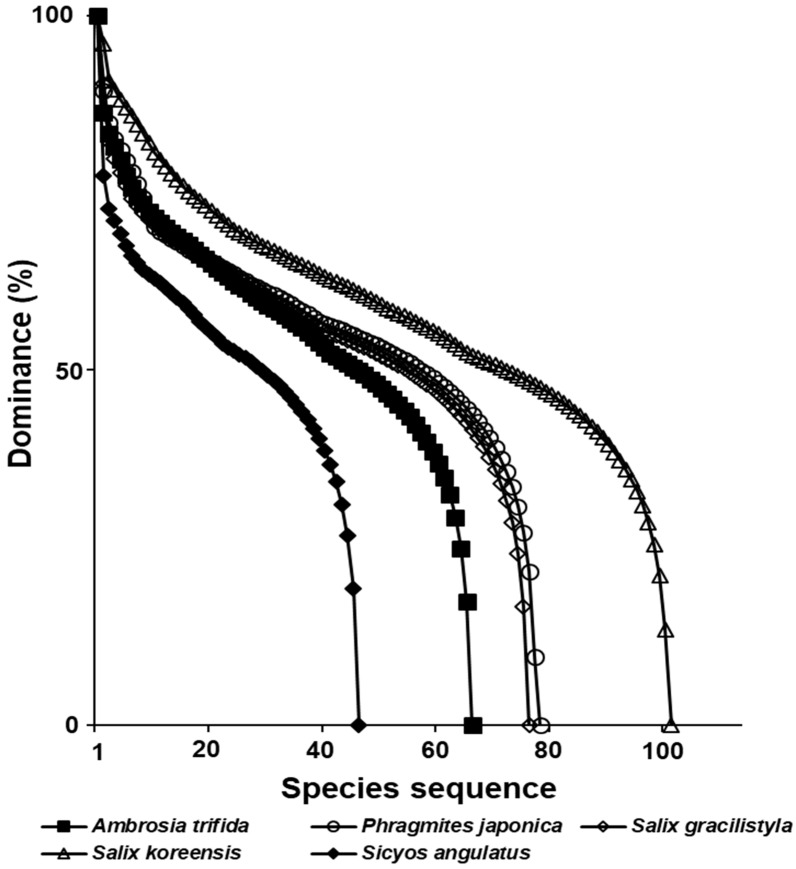
A comparison of species rank–abundance curves of two exotic species stands established in the riparian zone (*A. trifida* and *S. angulata*) and three native riparian vegetation stands (*S. pierotii*, *S. gracilistyla*, and *P. japonica*).

**Figure 15 biology-12-00826-f015:**
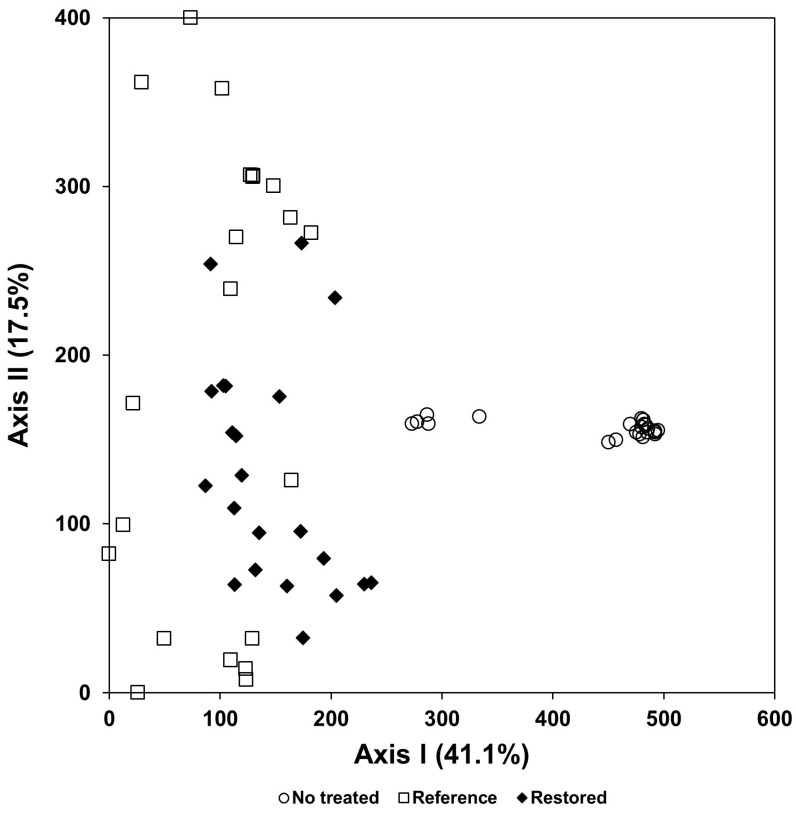
Stand ordination of the restored, non-restored, and reference stands. Non-restored stands were dominated by *A. trifida* or *S. angulate*, whereas the reference stands were dominated by *S*. *pierotii* and the restored sites were treated based on the reference information obtained from the natural river. Data of the reference stands were collected in the Suip stream, a natural stream which was left in its natural process for more than 70 years since the Korean War.

**Figure 16 biology-12-00826-f016:**
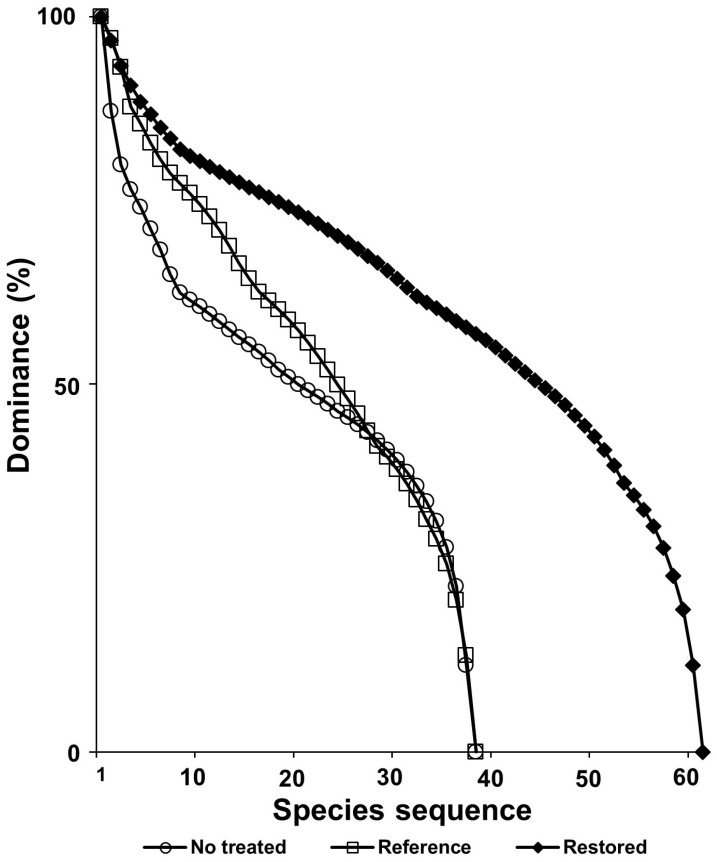
A comparison of species rank–abundance curves among the sites restored by introducing *S. pierotii* (Sp) and *S. gracilistyla* (Sg), nonrestored sites dominated by *A. trifida* (At) and *S. angulatus* (Sa), and reference sites.

## Data Availability

Raw exotic plant data is available on IKAS (https://kias.nie.re.kr/home/main/main.do accessed on 3 June 2023).

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
