# Peer review of "Distribution, Effect, and Control of Exotic Plants in Republic of Korea"

_biology, 2023, doi:10.3390/biology12060826_

Round 1

Reviewer 1 Report

The work by Soon Lim et al. entitled „Distribution, effect, and control of exotic plants in South Korea” presents the results of the study which try to answer to the questions which taxa invade native vegetation of South Korea, which ecosystems are most susceptible, how to manage described invasions and what are the ecological repercussions of the invasions.

The clarity of the work is very good, and I have only some minor suggestions which could improve the work a little.

The figure 1, maps could be rearranged to be made a little bigger to more comfortable analyze by the reader, (there is a free space in this place, so the rearrangement could allow to enlarge each map)

Figure 2. – the description of x axis is hard to decode because there are a lot of names under the bars and it is hard to say which name corresponds to which bar. I would rather use a number for each name, which could be decoded in the figure description,

Lines 248-255 – it should be mentioned whether the data met the assumptions of normality of distribution, which is needed for both Pearson`s correlation as well as ANOVA (and if the researchers checked that, and if yes, by which tests),

There is also a lack of information, which level of statistical significance was used in the statistical analyses in the whole work,

In figure 6 there should be information what depict the error bars on the graph, is it standard deviation, standard error or rather some other statistical measure?

Author Response

Response to Reviewer

Thank you for your kind advice and comments.

We faithfully revised our manuscript by referring to your valuable advice and comments faithfully answered your request as the follows.

Sincerely yours.

C.S. Lee

Comments and Suggestions for Authors

The work by Soon Lim et al. entitled „Distribution, effect, and control of exotic plants in South Korea” presents the results of the study which try to answer to the questions which taxa invade native vegetation of South Korea, which ecosystems are most susceptible, how to manage described invasions and what are the ecological repercussions of the invasions.

The clarity of the work is very good, and I have only some minor suggestions which could improve the work a little.

The figure 1, maps could be rearranged to be made a little bigger to more comfortable analyze by the reader, (there is a free space in this place, so the rearrangement could allow to enlarge each map)

☞ We increased the map’s size to make it easier for readers to see.

Figure 2. – the description of x axis is hard to decode because there are a lot of names under the bars and it is hard to say which name corresponds to which bar. I would rather use a number for each name, which could be decoded in the figure description,

☞ We revised Figure 2 by accepting the reviewer’s comment. Lines 292-93

Lines 248-255 – it should be mentioned whether the data met the assumptions of normality of distribution, which is needed for both Pearson`s correlation as well as ANOVA (and if the researchers checked that, and if yes, by which tests),

☞ Our data meet the assumptions of distribution normality required for both correlation analysis and ANOVA test and we added them. Lines 278-282

There is also a lack of information, which level of statistical significance was used in the statistical analyses in the whole work,

☞ We added the level of statistical significance by accepting the reviewer’s comment. Lines 278-282

In figure 6 there should be information what depict the error bars on the graph, is it standard deviation, standard error, or rather some other statistical measure?

☞ We added a description of the error bars. Lines 372-378

Reviewer 2 Report

The work is very interesting and necessary. However, I find the presentation of the results in the section "3.6 Spatial distribution of A. altissima as an exotic plant" a bit confusing. Figure 7 also seems unclear to me.

The inclusion of an appendix with the list of invasive alien species is necessary.

Author Response

Response to Reviewer

Thank you for your kind advice and comments.

We faithfully revised our manuscript by referring to your valuable advice and comments faithfully answered your request as the follows.

Sincerely yours.

C.S. Lee

The work is very interesting and necessary. However, I find the presentation of the results in the section "3.6 Spatial distribution of A. altissima as an exotic plant" a bit confusing. Figure 7 also seems unclear to me.

☞ A. altissima is distributed around the side of the hiking trail. Therefore, Figure 6 shows the state it occupies by evaluating it as a Braun-Blanquet cover class. And as a restorative treatment, a mantle vegetation was introduced at the edge of the forest and its changes were added.

☞ Figure 7 shows the spatial distribution of plant colonies dominated by exotic plants in Seoul. And the other map shows a spatial distribution of elevation. As you can see from the map, Seoul is a basin surrounded by mountains, and it can be seen that the plant communities dominated by exotic plants are mainly distributed in lowlands with low altitudes. These two maps were prepared to explain that more exotic plants are settling down because the lowlands have a stronger frequency and intensity of human interference than the uplands.

The inclusion of an appendix with the list of invasive alien species is necessary.

☞ We added a list of exotic plants investigated in South Korea as an Appendix table by accepting the reviewer’s comment. Line 364-366, 382, 668.

Reviewer 3 Report

In figure 2 Leguminosae are lacking?

Missing out seem figures 14-16?

Subsection on line 257 is superfluous

line 102: The cost of managing...

line 124: At .... levels not in levels (and many more such examples)

line 617: At the national level instead of in the level

line 630: Synthesizing instead of Synthesized

Author Response

Response to Reviewer 3

Thank you for your kind advice and comments.

We faithfully revised our manuscript by referring to your valuable advice and comments faithfully answered your request as the follows.

Sincerely yours.

C.S. Lee

Comments and Suggestions for Authors

In figure 2 Leguminosae are lacking?

☞ We revised figure 2 and the description.

Missing out seem figures 14-16?

☞ It is our mistake. We revised those Figure numbers.

Subsection on line 257 is superfluous

☞ We revised that part.

☞ We cut and trimmed this part by accepting reviewer’s comment.

Comments on the Quality of English Language

line 102: The cost of managing...

☞ We revised this part by accepting reviewer’s comment.

line 124: At .... levels not in levels (and many more such examples)

☞ We revised this part by accepting reviewer’s comment.

line 617: At the national level instead of in the level

☞ We revised this part by accepting reviewer’s comment.

line 630: Synthesizing instead of Synthesized

☞ We revised this part by accepting reviewer’s comment.

Reviewer 4 Report

Dear Editor and authors,

This research focused on the distribution, effect, and control of exotic plants in South Korea, which provides useful data for us to understand the distribution and management of exotic plants in North Korea. Although this paper is descriptive and lacks innovation, it provides a detailed analysis of the distribution of foreign invasive species in South Korea and is worthy of publication. However, I think there are still many issues with the writing of this paper, such as unclear methodology and chaotic numbering of figures. Here I have some major comments.

In the abstract, please do not use abbreviations, such as R5, it is difficult to understand.

Line 19-20: I cannot understand the “vegetation type richness”, and “ecological diversity”, please give a definition when it appeared first.

Line 24: Has the species composition of invasive alien species changed? The results here are not very clear.

Line 39: Remove “Oftentimes”.

Line 66: Too many references! Please match different references with the previous results one by one.

Line 92-101: These two paragraphs can be merged into one paragraph.

Line 178: “S. gracilistyla” instead of “Salix gracilistyla”. When a species name of the same genus appears for the second time, the genus can be abbreviated.

Line 198: Please give the summary methods, rather than just citing literature.

Line 204: What is the life form divided?

Line 220: “Q. mongolica” instead of “Q. Mongolica”?, and “A. altissima” instead of “Ailanthus altissima”.

Line 258: “subsection” instead of “Subsection”.

Line 258-261: I suggest the author can provide a list of exotic species in the appendix.

Line 275-278: What does the meaning of “R5”, “R3”, and “R2”, please give an explanation on these abbreviations in the method section.

Figure 6 (Line 314): Please give the P-value in the figure. Two Figure 6 occurred in the main text.

Line 318: This is Figure 4, and one Figure 4 also occurred in Lines 294-296.

Line 341-342, and another Figure 6: Please use the difference letter to show the significant difference between different traits in this figure.

Figure 7: Please explain the Cover class.

I think the language could be more concise. 

Author Response

Response to Reviewer

Thank you for your kind advice and comments.

We faithfully revised our manuscript by referring to your valuable advice and comments faithfully answered your request as the follows.

Sincerely yours.

C.S. Lee

Comments and Suggestions for Authors

Dear Editor and authors,

This research focused on the distribution, effect, and control of exotic plants in South Korea, which provides useful data for us to understand the distribution and management of exotic plants in North Korea. Although this paper is descriptive and lacks innovation, it provides a detailed analysis of the distribution of foreign invasive species in South Korea and is worthy of publication. However, I think there are still many issues with the writing of this paper, such as unclear methodology and chaotic numbering of figures. Here I have some major comments.

In the abstract, please do not use abbreviations, such as R5, it is difficult to understand.

☞ We revised this part by accepting the reviewer’s comment. Line 28

Line 19-20: I cannot understand the “vegetation type richness”, and “ecological diversity”, please give a definition when it appeared first.

☞ We added the definition of the term by accepting the reviewer’s comment. Lines 35-36

Line 24: Has the species composition of invasive alien species changed? The results here are not very clear.

☞ It means that species composition of the vegetation infected by exotic species.

Line 39: Remove “Oftentimes”.

☞ We deleted the word by accepting the reviewer’s comment. Line 55

Line 66: Too many references! Please match different references with the previous results one by one.

☞ We revised this part by accepting the reviewer’s comment. Lines 75-82

Line 92-101: These two paragraphs can be merged into one paragraph.

☞ We revised this part by accepting the reviewer’s comment. Lines 108-116

Line 178: “S. gracilistyla” instead of “Salix gracilistyla”. When a species name of the same genus appears for the second time, the genus can be abbreviated.

☞ We revised this part by accepting the reviewer’s comment. Line 194

Line 198: Please give the summary methods, rather than just citing literature.

☞ We explained the survey methods by accepting the reviewer’s comment. Lines 217-224

Line 204: What is the life form divided?

☞ We divided life forms based on dormancy form, longevity, disseminule form, growth form, and radicoid form. Lines 225-236

Line 220: “Q. mongolica” instead of “Q. Mongolica”?, and “A. altissima” instead of “Ailanthus altissima”.

☞ We revised this part by accepting the reviewer’s comment. Line 251

Line 258: “subsection” instead of “Subsection”.

☞ We deleted the word.

Line 258-261: I suggest the author can provide a list of exotic species in the appendix.

☞ We added a list of exotic species as the appendix table. Line 672

Line 275-278: What does the meaning of “R5”, “R3”, and “R2”, please give an explanation on these abbreviations in the method section.

☞ We revised this part by accepting the reviewer’s comment. Lines 225-236

Figure 6 (Line 314): Please give the P-value in the figure. Two Figure 6 occurred in the main text.

☞ We revised this part by accepting the reviewer’s comment. Lines 278-282, 345

Line 318: This is Figure 4, and one Figure 4 also occurred in Lines 294-296.

☞ It is our mistake. We revised this part by accepting the reviewer’s comment.

Line 341-342, and another Figure 6: Please use the difference letter to show the significant difference between different traits in this figure.

☞ We revised this part by accepting the reviewer’s comment. Line 371

Figure 7: Please explain the Cover class.

☞ We explained that it is the cover class of an exotic plant, Ageratina altissima, measured by the Braun-Blanquet scale in Methods section.

Comments on the Quality of English Language

I think the language could be more concise. 

☞ We refined the sentences of the manuscript. If more is needed in the future, it will be reviewed by native speakers.
